# A concise and scalable chemoenzymatic synthesis of prostaglandins

Yunpeng Yin [1,3], Jinxin Wang [2,3] & Jian Li [1] ✉

Prostaglandins have garnered significant attention from synthetic chemists due to their exceptional biological activities. In this report, we present a concise chemoenzymatic synthesis method for several representative prostaglandins, achieved in 5 to 7 steps. Notably, the common intermediate bromohydrin, a radical equivalent of Corey lactone, is chemoenzymatically synthesized in only two steps, which allows us to complete the synthesis of prostaglandin $F_{2\alpha}$ in five steps on a 10-gram scale. The chiral cyclopentane core is introduced with high enantioselectivity, while the lipid chains are sequentially incorporated through a cost-effective process involving bromohydrin formation, nickel-catalyzed cross-couplings, and Wittig reactions. This cost-efficient synthesis route for prostaglandins holds the potential to make prostaglandin-related drugs more affordable and facilitate easier access to their analogues.

Natural prostaglandins (PGs) are a family of lipid compounds generated from arachidonic acid by cyclooxygenase (COX)[1]. Due to the unique biological properties and the medicinal value of prostaglandins (PGs), they have gained extensive attention from medicinal chemists. However, natural prostaglandin molecules often have poor chemical stability, rapid in vivo metabolism, and are frequently associated with corresponding side effects, posing challenges for direct medical use[2]. In recent years, medicinal chemists have developed a series of structurally diverse prostaglandin derivatives and analogs based on this molecular framework to attempt to address these issues (Fig. 1). More than 20 prostaglandin-class drugs have been successfully approved for the treatment of various diseases, with global annual sales amounting to billions of dollars, indicating broad demand and a huge market. Therefore, developing concise and efficient routes for synthesizing prostaglandins and related drugs is of great significance in meeting market demands, reducing medication costs, and facilitating new drug development.

In the decades following the landmark synthesis of prostaglandin $F_{2\alpha}$ by the Corey laboratory[3–10], synthetic chemists have developed a number of efficient synthetic approaches using Corey lactone as a key intermediate for the synthesis of various prostaglandins and their analogues[11–24]. Chen research group has previously accomplished the shortest synthesis route for Corey lactone[23,24], achieving the asymmetric synthesis of Corey lactone in just four steps starting from commercially available starting materials. A collection of highly effective synthetic methodologies, developed by more than 20 research teams from prestigious laboratories, including those led by Stork, Woodward, Nicolaou, Danishefsky, Carreira, Noyori, Aggarwal, and Zhang have been established for crafting novel synthesis pathways for prostaglandins and their derivatives[25–57]. These approaches notably bypass the use of the traditional Corey lactone intermediates. There are two strategies that are representative in terms of efficiency: the Zhang lab employed several elegant noble metal catalytic reactions, such as iridium, rhodium, palladium, ruthenium, to accomplish the scalable synthesis of prostaglandins[47]. On the other hand, the Aggarwal lab utilized an efficient organocatalytic chemistry and completed the synthesis of prostaglandins in only seven steps[33].In these synthetic reports, the radical cleavage strategy by the Baran group[22] and the Baeyer–Villiger reaction-based strategy by the Chen group[23,24] are very inspirational to the retrosynthetic analysis of this work. Due to Chen lab's use of a dichloro precursor as their substrate, their synthesis process totaled eight steps to reach prostaglandin $F_{2\alpha}$. Meanwhile,

[1]Frontiers Science Center for Transformative Molecules, School of Chemistry and Chemical Engineering, Shanghai Key Laboratory for Molecular Engineering of Chiral Drugs and Zhangjiang Institute for Advanced Study, Shanghai Jiao Tong University, Shanghai, China. [2]Department of Phytochemistry, School of Pharmacy, Second Military Medical University, Shanghai, China. [3]These authors contributed equally: Yunpeng Yin, Jinxin Wang.
✉e-mail: jianlizcz@sjtu.edu.cn

**Fig. 1 | Prostaglandins and their biosynthesis via cyclooxygenation. a** Some representative prostaglandins. **b** Generation of Prostaglandin G2 from arachidonic acid by cyclooxygenase.

Baran lab opted for the more expensive Corey lactone, which requires at least four steps to synthesize from a cost-efficient starting material. In the past decade, chemists have developed many excellent conditions for radical reactions for C–C bond formation, making radical disconnection an important complement to its polar and pericyclic counterparts in retrosynthetic logic[58]. In recent years, chemoenzymatic strategies have also been employed by organic chemists for the synthesis of complex molecules[59]. Renata lab has pioneered the merger of chemoenzymatic and radical-based retrosynthetic logic, and simplified the synthesis of meroterpenoid natural products[60]. Thereby, we believe that there is room for efficiency improvement in the synthesis of prostaglandins through a design that combines both free radical approach and enzymatic methods.

In our retrosynthetic analysis of prostaglandins (Fig. 2a), the cis double bonds in prostaglandins would be installed using classic Wittig reaction, a polar disconnection. Next, we conceived a nickel-catalyzed reductive coupling, a radical approach, for the installation of the trans double bond side chains. This disconnection entailed that bromohydrin **8** would be used as a radical version of Corey lactone as a common intermediate for prostaglandins. In turn, **8** could be obtained by simply treating chiral lactone **9** with NBS and water. For the synthesis of chiral lactone **9**, two synthetic strategies were devised. The first one involved an in vivo enzymatic Baeyer–Villiger oxidation in an enantiomer divergent fashion. Although the strategy enables the one-step preparation of lactone **9**, the in vivo biotransformation reaction is more challenging to achieve in traditional chemistry laboratories or medicinal chemistry laboratories. Therefore, we developed a supplementary strategy that utilizes commercially available lipase to prepare chiral compound **11**, followed by a Johnson–Claisen rearrangement reaction to obtain the lactone **9**. Based on our retrosynthetic analysis, the synthesis for Corey lactone equivalent **8** can be achieved in just two steps. Meanwhile, by utilizing free radical chemistry, the entirety of the trans-olefin side chains can be installed in one step, which would result in an overall reduction in total step count (Fig. 2b). Here, we report a concise and scalable chemoenzymatic synthesis of prostaglandins by merging chemoenzymatic and radical-based retrosynthetic logic.

## Results
### Two methods for the preparation of chiral lactone 9
To synthesize the required chiral lactone compounds (Fig. 3), a Johnson–Claisen rearrangement strategy was first developed. The achiral diol **12** can be obtained commercially at a price of $13 per gram. The asymmetric synthesis of **11** has been achieved through lipase-mediated desymmetrization[61], resulting in mono-acetate product **11**

with an enantiomeric excess of 95%. The mono-acetate product **11** can be transformed into the Johnson–Claisen rearrangement product **13** by using triethyl orthoacetate as the solvent and catalyzed by o-nitrophenol. The resulting product, without the need for separation, was directly treated with $K_2CO_3$ and MeOH to obtain lactone **9** in one pot. Compared to similar strategies in the literature, our modification reduced the number of steps by half[62]. Although this strategy can yield 13.2 g of lactone **9** in one run and performed in any chemistry laboratory, the expensive starting materials, and the need for prolonged high-temperature heating in the Johnson–Claisen rearrangement step make it difficult to further scale-up. Therefore, an enzymatic oxidative resolution was explored method using a more affordable racemic cyclobutanone **10**, available at a price of $2.3 per gram. Several enzymatic Baeyer–Villiger reactions[63–65] of cyclobutanone **10** have been reported, including a scale-up of this reaction to 2.5 g of lactone products per kilogram of cell broth[66]. In our preliminary study, the incorporation of an NADPH regeneration system increased the concentration of the products in the cell broth to 9.3 g/L. We opted to construct an *E. coli* strain that co-expresses glucose dehydrogenase with CHMO. In this system, CHMO$_{rhodo1}$ achieved a product concentration of 32 mM. In subsequent optimizations, it was discovered that the capacity of glucose dehydrogenase to regenerate NADPH was significantly lower than the efficiency of CHMO$_{rhodo1}$ in our reaction system. Upon switching to the phosphite dehydrogenase Opt-13 developed by the Zhao group[67], complete conversion of cyclobutanone to the product was achieved at concentrations of 40 mM and 83 mM, respectively. Control experiments demonstrated that as reaction concentrations continued to rise, the reaction activity began to decrease sharply due to the influence of high concentration of sodium phosphite. Therefore, while scaling up, the reaction can be run at 83 mM with full conversion, allowing the preparation of over 100 g of lactone product **9**. The ee values of the lactone compounds **9** and **14** obtained in the various NADPH regeneration systems and reactions of different scales mentioned above remained essentially same: 95% ee for lactone **9** and 97% ee for lactone **14**.

### Bromohydrin formation and radical connection of ω-chains
The formation of bromohydrin **8** proved to be more challenging than expected. In the literature, the reaction of the same substrate with NBS-water system has been studied, showing that the reaction typically affords the undesired bromohydrin diastereomer[68]. Inspired by refs. [69–72], DMSO was used as a Lewis base to alter the behavior of the bromonium ion. In the presence of DMSO, a reversal in the direction of bromonium formation was observed. Upon further screening of other

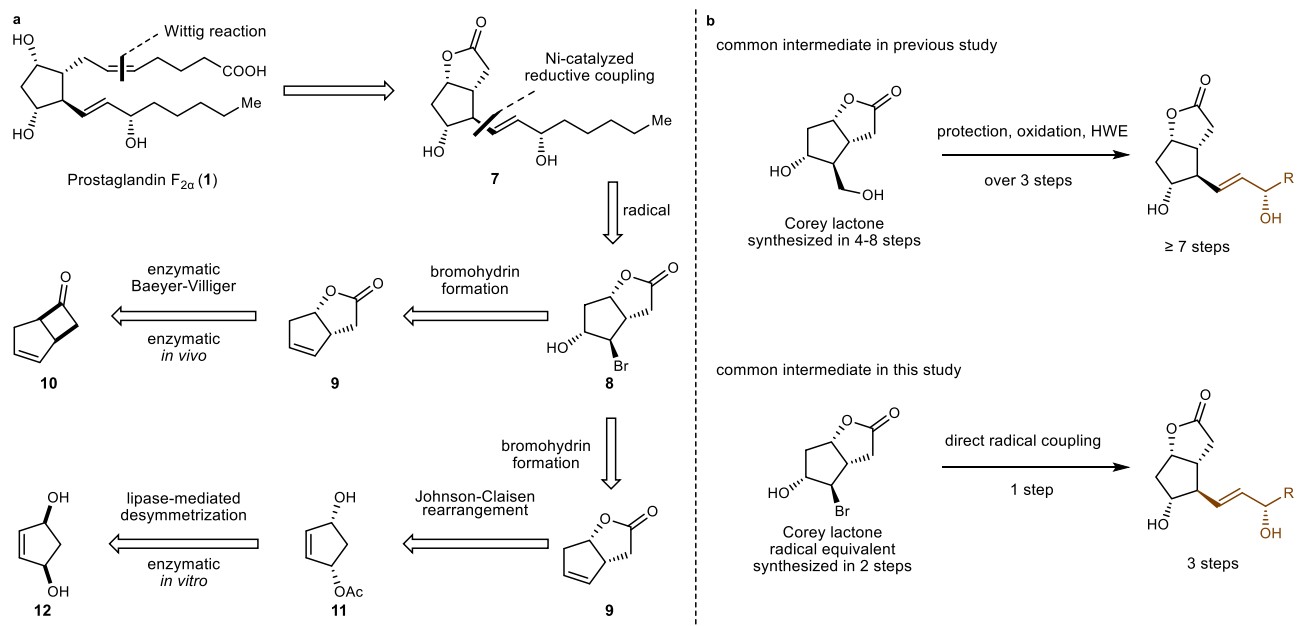

**Fig. 2 | Our retrosynthetic analysis and comparison with previous studies. a** Our retrosynthetic analysis of prostaglandin F$_{2\alpha}$. **b** Comparison with previous studies using Corey latone and Corey latone radical equivalent.

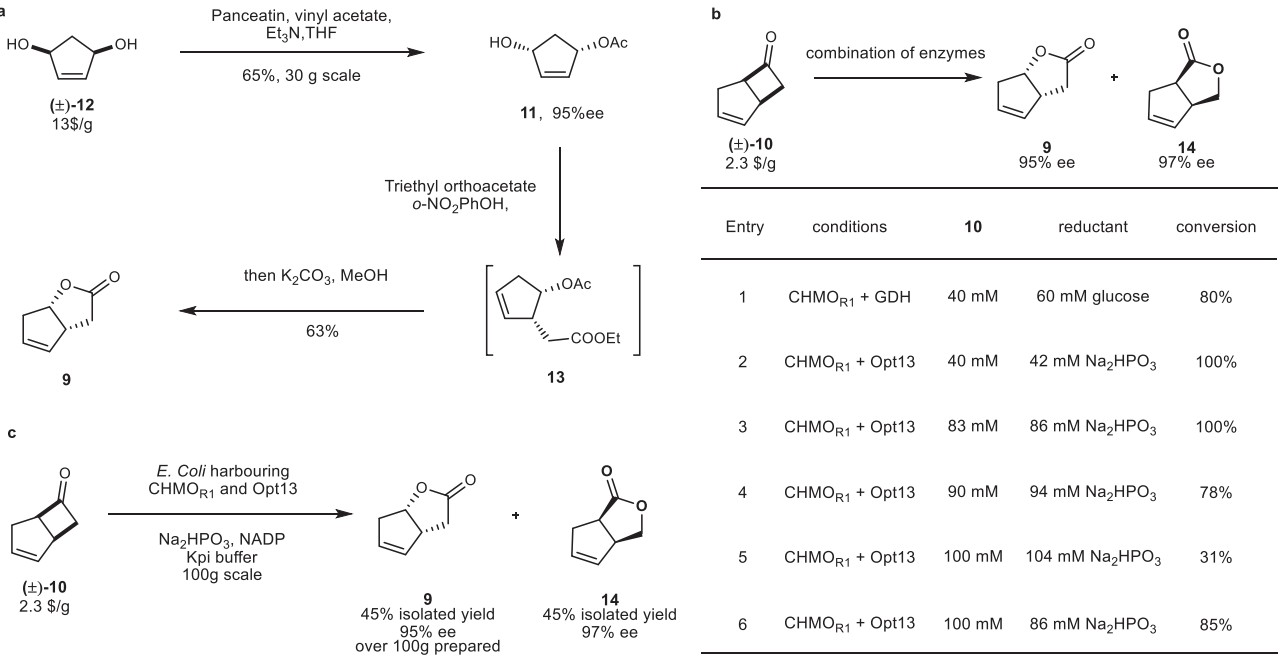

**Fig. 3 | in vivo- and in vitro- chemoenzymatic synthesis of lactone 9. a** Synthesis of lactone **9** based on an in vitro enzymatic desymmetrization and Johnson–Claisen reaction. **b** Matching NADPH regenerating system with CHMO$_{rhodo1}$. **c** Scale-up of enzymatic Baeyer−Villiger reaction. THF tetrahydrofuran, NADP nicotinamide adenine dinucleotide phosphate.

co-solvent system, the reaction's overall yield and the ratio of our desired product reached their highest levels when DMSO was used as a co-solvent with chloroform (Fig. 4a). With bromide **8** in hand, the radical coupling reaction of the first sidechain was tested. Since the pioneering work of the Weix research group[73–75], a wide range of conditions for nickel-catalyzed reductive coupling reactions have been reported[76]. Some examples showed direct involvement of bromohydrins in the coupling reaction[77]. The condition reported by the Gong group was attempted[77], which fortunately afforded the coupling product in 49% yield (Fig. 4b). After analyzing the composition of the

products, we found that due to the weak alkaline nature of the coupling conditions, one of the main by-products is compound **17**, a product of epoxide ring closure. Catalytic amount of bidentate ligand **18** were employed as a replacement for the equivalent use of pyridine; however, the reaction produced more epoxide. To fundamentally suppress the formation of epoxides, in Entry 3, 1.1 equivalents of N-(Trimethylsilyl)imidazole was added to protect the hydroxyl group in situ. We anticipate that the one equivalent of imidazole generated in situ, as a product of the temporary protection of hydroxyl groups by N-(Trimethylsilyl)imidazole, can act as a ligand in the reaction. Entry 3

**Fig. 4 | Efficient synthesis of key intermediate 8 and ω-chains installation.**
**a** Screening the optimal condition for bromohydrin **8** formation reaction.
**b** Screening the optimal condition for reductive coupling reaction of bromohydrin
**8**. **c** Bidentate ligands used in this work. **d** Reaction scope of reductive coupling
reaction of bromohydrin **8**. NBS N-bromosuccinimide, DMSO dimethyl sulfoxide,
DMA dimethylacetamide, TSIM N-(trimethylsilyl)imidazole.

provided the coupling product in 52% yield along with 37% yield of the
elimination product **9**. To our delight, under this condition no epox-
idation formation was found. Considering that imidazole may not
serve as a good ligand, upon adding an equivalent amount of pyridine
or 15% of a bidentate ligand **18** together with N-(Trimethylsilyl)imida-
zole, the reaction yield increased to above 83%. The substrate scope
was then expanded based on the optimal conditions, allowing the
installation of three different side chains of prostaglandins with high
yields. Due to the cost consideration that ligand prices are significantly
higher than the nickel source in nickel-catalyzed reactions, we opted
for coupling conditions with an equivalent amount of pyridine for
subsequent reaction scale-up. In addition, coupling reactions with
aromatic and heterocyclic substrates were attempted, which also
resulted in a high yield of the coupled products using ligand **19**.

### Completing the syntheses of prostaglandins
After the optimization of chiral skeleton construction and key radical
coupling reaction, the remaining steps were relatively straightforward.
During the scale-up process, the yield of the bromohydroxylation
reaction and nickel-catalyzed reductive coupling reactions remained
consistent. Furthermore, through recrystallization bromohydrin **8** was
obtained with a yield of 86% and an enantiomeric excess of greater
than 99%. The nickel-catalyzed reductive coupling reactions exhibited
a decreased yield of 73% at a 10-g scale, allowing us to obtain 12 g of
product **7** in a single batch. After obtaining compound **7**, some
improvements were made to known methods and completed the 10-g
scale synthesis of prostaglandin $F_{2\alpha}$ through lactone reduction and
Wittig reaction (Fig. 5a). It is worth noting that we synthesized 10.6 g of
prostaglandin $F_{2\alpha}$ using only 14.2 g of lactone compound **9**. Using a
similar strategy, we finished the synthesis of 2.13 g of fluprostenol,
1.89 g of bimatoprost, 1.82 g of cloprostenol, and 1.29 g of latanoprost
(Fig. 5b, c), each of the synthesis is initiated with 1.6 g of lactone

compound **9**. For the synthesis of latanoprost, to avoid the use of
noble metals, Raney nickel was used as a catalyst and obtained the
product of double bond hydrogenation in 99% yield.

### Discussion
In summary, We have achieved a scalable synthesis of prostaglandins
with high enantioselectivity using cost-effective commercial starting
materials. Our synthesis of prostaglandins represents one of the
shortest route reported to date, among which the synthesis of pros-
taglandin $F_{2\alpha}$ was accomplished in just five steps. Two distinct meth-
odologies for the synthesis of chiral lactone **9** were developed: (1) a
Johnson–Claisen strategy, which is readily feasible in any chemistry
laboratory, and (2) an enzymatic oxidative resolution strategy that is
more scalable and cost-effective. Furthermore, by incorporating a
radical-based strategic bond disconnection, we can divergently syn-
thesize various prostaglandin drugs through nickel-catalyzed reduc-
tive couplings and olefination reactions. The route has high industrial
application value due to its cost-effectiveness and the fact that it does
not utilize any noble metals.

### Methods
#### General procedure for biocatalytic oxidation with recombinant *E. coli*
An overnight culture of *E. coli* BL21(DE3) cells harboring pET-22b(+)-
based vector for expressing the appropriate CHMO$_{rhodo1}$ and pRSF-
Opt-13 plasmid was used to inoculate 500 mL TB media (in 2 L Erlen-
meyer flask) containing 50 μg/mL kanamycin, 50 μg/mL ampicillin. The
cultures were shaken at 250 rpm at 37 °C until an optical density of
OD600 = 0.7–1.0 was reached. The cultures were cooled on ice for
20 min and then induced with riboflavin and IPTG to final concentra-
tions of 1.0 μM and 0.5 mM, respectively. The cultures were shaken at
150 rpm at 20 °C for a further 20 h. Cells were harvested by

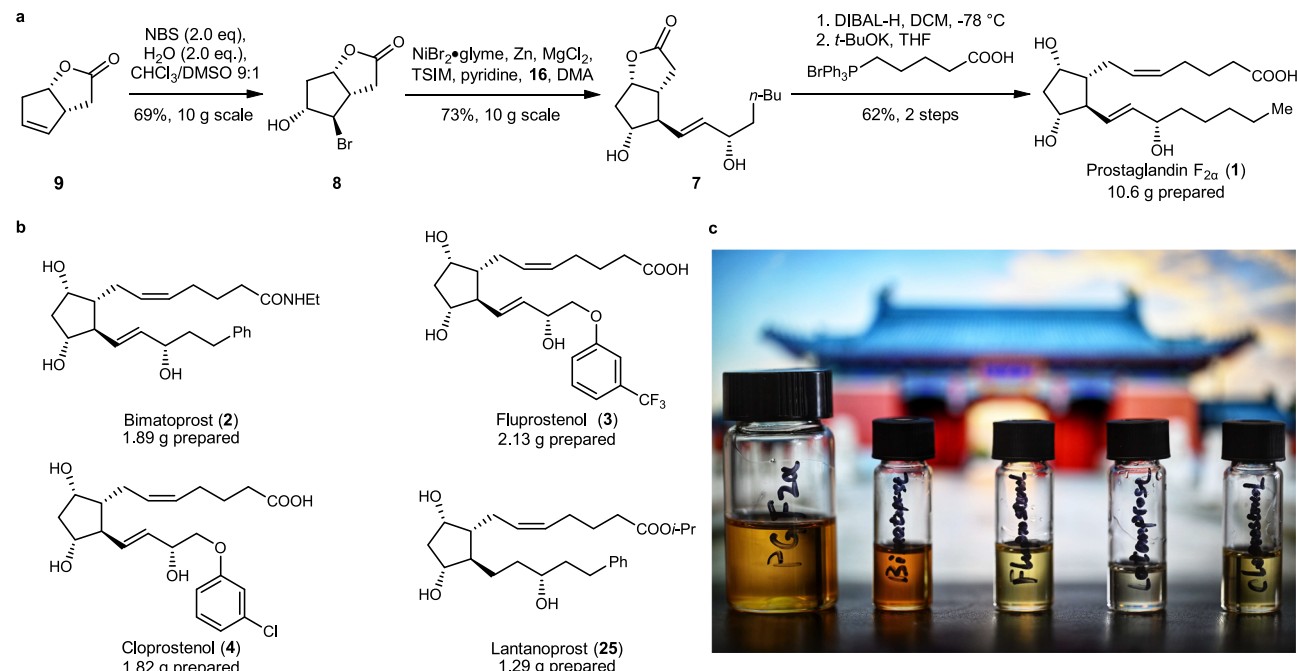

**Fig. 5 | Completion of the chemoenzymatic synthesis of prostaglandins. a** A decagram scale synthesis of prostaglandin $F_{2\alpha}$. **b** A gram-scale synthesis of related prostaglandins. **c** A photo of prostaglandins prepared in one batch; this photo was taken by Jian Li (background: the "temple gate" of SJTU).

centrifugation (4 °C, 15 min, 4121 g), and resuspended in 500 mL kPi buffer (50 mM, pH = 8.00) to an $OD_{600}$ = 10 into a 2 L Erlenmeyer flask. To the mixture was sequentially added a pre-dissolved solution of 4.5 g ketone **10** in 25 mL DMSO, $Na_2NADP \cdot 4H_2O$ (654 mg, 0.83 mmol), $Na_2HPO_3 \cdot 5H_2O$ (9.3 g, 43.0 mmol). The Erlenmeyer flask was shaken at 150 rpm at 25 °C for 20 h. The mixture was extracted with EtOAc (300 mL × 3), and the combined organic extracts were concentrated in vacuo gives 1:1 mixture of lactone **9** and **14**.

## Data availability

Crystallographic data for the structure reported in this article have been deposited at the Cambridge Crystallographic Data Centre, under deposition nos. CCDC 2312333 (**8**) and CCDC 2319091 (**21**). Copies of the data can be obtained free of charge via https://www.ccdc.cam.ac.uk/structures. All other characterization data and detailed experimental procedures are available in the supplementary materials.

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

## Acknowledgements

The authors thank Professor Hans Renata and Professor Ang Li for valuable suggestions and helpful discussions. This work was supported by the National Key R&D Program of China (grant no. 2023YFA1506700 to J.L.), The Shanghai Science and Technology Development Funds (grant no. 22QA1404100 to J.L.), and Fundamental Research Funds for the Central Universities (23×010301599 to J.L.). The authors thank the Prof. Shuangjun Lin for generous access to their lab space and instrumentations.

## Author contributions

J.L. conceived this project. Y.Y., J.W., and J.L. performed the experiment. J.L. and Y.Y. co-wrote the manuscript. All authors discussed the results and commented on the manuscript.

## Competing interests

The authors declare no competing interests.
