## [Peer Review File · Nature Communications]

A Concise and Scalable Chemoenzymatic Synthesis of ProstaglandinsREVIEWER COMMENTS

Reviewer #1 (Remarks to the Author):

This paper describes a concise chemoenzymatic synthesis of several representative prostaglandins.

The first route employs a lipase-mediated desymmetrization of the diol giving the mono-acetate product which is then transformed into the Johnson-Claisen rearrangement product by using triethyl orthoacetate as the solvent. No reference is provided but the Johnson-Claisen rearrangement step was done on the same substrate by Schneider giving the same product (<https://pubs.rsc.org/en/content/articlepdf/1986/c3/c39860001298>)

The second approach uses an enzyme catalysed resolution of a chiral but racemic cyclobutanone. Again, this is known chemistry and has been scaled up too. Although it is stated that “allowing the preparation of over 100 grams of lactone product 9” the experimental only details the production of 2.32 g. This used 0.5L of one reagent, so to scale it up 50 fold would be a considerable undertaking. Details for the synthesis of >100g should be given. Also, I would not be able to follow the experimental procedure as described, it is too sparse. The two lactones are expected to be difficult to separate (as indicated here - *Angew. Chem. Int. Ed.* 2019, 58, 9923) and the authors should give more information on this. I am not convinced that this is the best route since the dichlorocyclobutanone must be a precursor to the cyclobutanone itself and the dichlorocyclobutanone undergoes the BV reaction very effectively (*Angew. Chem. Int. Ed.* 2019, 58, 9923) with an easy separation of lactones. Some of this needs to be included in the paper.

The next step was formation of the bromohydrin from the alkene where it was found that using CHCl₃/DMSO mixture gave optimum selectivity.

I did like the next step, the application of Weix's Ni-catalysed coupling of the alkyl bromide with the alkenyl halide to bring in the lipophilic side chain. This key step was used to prepare a small set of PGs showing its potential.

Overall, this is quite a good paper that could be accepted in Nature Communications particularly because the Ni-catalysed coupling enables a range of PGs to be made easily and on scale.

Other issues:

Citation of Aggarwal's work misses the following references:

Org. Synth., 2022, 99, 139-158.

Chem. Rec., 2020, 20, 936–947

ACS Cent. Sci., 2020, 6, 995–1000.

The hazards associated with the PG products should be highlighted since they are all highly active products. What precautions did the authors take?

Reviewer #2 (Remarks to the Author):

Prostaglandins are important molecules in chemistry and biology due to their unique structural features, chemical reactivities, and biological activities. Many synthetic routes have been reported for the construction of these molecules. This manuscript describes a new chemoenzymatic synthetic route for prostaglandin F₂α and its four analogues. The syntheses were scalable and efficient, demonstrating advancement in both enzymology and chemical methodologies for molecule construction. Therefore, the synthetic route possesses high academic value and industrial potentials. On the other hand, supplementary materials highlight the meticulous nature of the work, evident in the purity of the ¹H and ¹³C NMR spectra.

Overall, this reviewer recommends the publication of this paper in Nature Communications.

Two changes should be made before the publication:

1. Compared to similar strategy in the literature, authors reported a Johnson-Claisen reduced the number of steps by half. Which is a big improvement, similar strategy should be cited in the text: J. Chem. Soc., Chem. Commun. 1986, 1298.
2. In the last paragraph, “Two distinct methodologies for the synthesis of chiral lactone 7 were developed” chiral lactone 7 should be chiral lactone 9.

Reviewer #3 (Remarks to the Author):

Prostaglandins are important hormone-like lipid compounds with unique structures and a broad range of bioactivities, including inflammatory responses, blood clotting, vasodilation,

reproduction functions. Consequently, the efficient synthesis of such molecules command significant attention from synthetic chemists. In this manuscript, the author has accomplished a route that combines enzymatic and chemical methods, realizing one of the most concise synthetic pathways for this category of molecules to date. As a merger of chemoenzymatic and recent radical-based chemical methodologies, the synthetic route has high academic value. The cost-effective and the noble-metal-free nature of this synthesis confer a high potential for industrial application. In the supplementary information, pure NMR spectra demonstrate the reaction's efficiency and the author's meticulousness from another perspective. The reviewer believes that this work is suitable for publication in Nature Communications.

The suggested revisions are as follows:

1. The authors should double check the numbering of compounds. For example, in the end of the text: lactone 9 was mislabeled as lactone 7.
2. In figure 2 retro analysis, the word “polar” seems to be meaningless, should it be removed?
3. To make the text more accessible to a broader readership, uncommon abbreviations such as TSIM should have their full terms clarified in the caption.

Point-by-point Response Reviewers' Comments

Reviewer #1 (Remarks to the Author):

This paper describes a concise chemoenzymatic synthesis of several representative prostaglandins.

The first route employs a lipase-mediated desymmetrization of the diol giving the mono-acetate product which is then transformed into the Johnson-Claisen rearrangement product by using triethyl orthoacetate as the solvent. No reference is provided but the Johnson-Claisen rearrangement step was done on the same substrate by Schneider giving the same product (<https://pubs.rsc.org/en/content/articlepdf/1986/c3/c39860001298>)

Reference added in the revised version.

The second approach uses an enzyme catalysed resolution of a chiral but racemic cyclobutanone. Again, this is known chemistry and has been scaled up too. Although it is stated that "allowing the preparation of over 100 grams of lactone product 9" the experimental only details the production of 2.32 g. This used 0.5L of one reagent, so to scale it up 50 fold would be a considerable undertaking. Details for the synthesis of >100g should be given. Also, I would not be able to follow the experimental procedure as described, it is too sparse.

Due to our oversight, in the methods section and supplementary information we only described the process for a single conical flask reaction; however, in actual fermentation operations, we typically conduct reactions in parallel with 12 or 24 Erlenmeyer flasks. Therefore, in practice, by combining multiple reactions, we are able to achieve a batch input of over 100 grams and obtain approximately 50 grams of product. The specific experimental details have been supplemented in the Supplementary Information.

The image below shows the specific details of the shaker used for fermentation. Due to the limitations of the shaker in the laboratory, we can only use 2L Erlenmeyer flasks for parallel experiments.

The two lactones are expected to be difficult to separate (as indicated here - Angew. Chem.

Int. Ed. 2019, 58, 9923) and the authors should give more information on this. I am not convinced that this is the best route since the dichlorocyclobutanone must be a precursor to the cyclobutanone itself and the dichlorocyclobutanone undergoes the BV reaction very effectively (Angew. Chem. Int. Ed. 2019, 58, 9923) with an easy separation of lactones. Some of this needs to be included in the paper.

The separation of the two lactone compounds does indeed pose certain challenges, but it is not entirely unfeasible. In our experiments, lactone mixtures can be separated on a larger-sized chromatography column. Below is an image showing the separation of the two lactone compounds on TLC (1 run):

As the later report by Chen's research group (Chem. Sci. 2021, 12, 10362.), when using dichlorocyclobutanone as the substrate at a concentration of 10 mmol/L, the yield is 38%. The dechlorinated lactone has been optimized by us to a 45% yield at a higher concentration of 83 mmol/L (10-fold of the product concentration). With lower substrate concentrations, the process of extracting large quantities of fermentation broth would also be more cumbersome. Therefore, after weighing the difficulties of scaling up against the challenges of separation, we have chosen our current route. We added the details on the work-up and the separation of this step in the Supplementary Information.

The next step was formation of the bromohydrin from the alkene where it was found that using CHCl₃/DMSO mixture gave optimum selectivity.

I did like the next step, the application of Weix's Ni-catalysed coupling of the alkyl bromide with the alkenyl halide to bring in the lipophilic side chain. This key step was used to prepare a small set of PGs showing its potential.

Overall, this is quite a good paper that could be accepted in Nature Communications particularly because the Ni-catalysed coupling enables a range of PGs to be made easily and on scale.

Other issues:

Citation of Aggarwal's work misses the following references:

Org. Synth., 2022, 99, 139-158.

Chem. Rec., 2020, 20, 936–947

ACS Cent. Sci., 2020, 6, 995–1000.

Reference added in the revised version (ref 38-40), along with a missed work from Chen lab(ref 41).

The hazards associated with the PG products should be highlighted since they are all highly active products. What precautions did the authors take?

All intermediate and final product reactions and post-treatments are strictly conducted in a fume hood, and all reaction waste are also collected and treated. The intermediates and final products are properly stored in a low-temperature refrigerator. It is worth mentioning that, as a total synthesis research group, we handle a variety of molecules with unknown activity every day. The potential hazards of these molecules could be more serious than known drugs, hence we take personal protective measures when handling compounds on a daily basis.

Reviewer #2 (Remarks to the Author):

Prostaglandins are important molecules in chemistry and biology due to their unique structural features, chemical reactivities, and biological activities. Many synthetic routes have been reported for the construction of these molecules. This manuscript describes a new chemoenzymatic synthetic route for prostaglandin F2 α and its four analogues. The syntheses were scalable and efficient, demonstrating advancement in both enzymology and chemical methodologies for molecule construction. Therefore, the synthetic route possesses high academic value and industrial potentials. On the other hand, supplementary materials highlight the meticulous nature of the work, evident in the purity of the ^1H and ^{13}C NMR spectra.

Overall, this reviewer recommends the publication of this paper in Nature Communications.

Two changes should made before the publication:

*1. Compared to similar strategy in the literature, authors reported a Johnson-Claisen reduced the number of steps by half. Which is a big improvement, similar strategy should be cited in the text: *J. Chem. Soc., Chem. Commun.* 1986, 1298.*

Reference added in the revised version.

2. In the last paragraph, "Two distinct methodologies for the synthesis of chiral lactone 7 were developed" chiral lactone 7 should be chiral lactone 9.

The incorrectly labeled compound numbers have been corrected.

Reviewer #3 (Remarks to the Author):

Prostaglandins are important hormone-like lipid compounds with unique structures and a broad range of bioactivities, including inflammatory responses, blood clotting, vasodilation, reproduction functions. Consequently, the efficient synthesis of such molecules command significant attention from synthetic chemists. In this manuscript, the author has accomplished a route that combines enzymatic and chemical methods, realizing one of the most concise synthetic pathways for this category of molecules to date. As a merger of chemoenzymatic and recent radical-based chemical methodologies, the synthetic route has high academic value. The cost-effective and the noble-metal-free nature of this synthesis confer a high potential for industrial application. In the supplementary information, pure NMR spectra demonstrate the reaction's efficiency and the author's meticulousness from another perspective. The reviewer believes that this work is suitable for publication in Nature Communications.

The suggested revisions are as follows:

1. The authors should double check the numbering of compounds. For example, in the end of the text: lactone 9 was mislabeled as lactone 7.

The incorrectly labeled compound numbers have been corrected.

2. In figure 2 retro analysis, the word "polar" seems to be meaningless, should it be removed?

The word "polar" was removed from figure 2.

3. To make the text more accessible to a broader readership, uncommon abbreviations such as TSIM should have their full terms clarified in the caption.

Authors updated the full terms of rare used abbreviations.

REVIEWER COMMENTS

Reviewer #1 (Remarks to the Author):

The authors have not provided sufficient details to show how the lactone isomers can be separated on scale. They are VERY close running and to separate >100g of crude material is a VERY challenging task. I urge the authors to show the TLCs of the lactone mixture and the column fractions that were collected and to give more details of how they do the column - dimensions of column, eluent/gradient, flow rate, size of fractions, etc.

The Johnson-Claisen rearrangement reference (62) should be given where the Johnson-Claisen rearrangement is discussed, not later in the paper.

After that, accept paper. Rest is fine.

Point-by-point Response Reviewers' Comments

Reviewer #1 (Remarks to the Author):

The authors have not provided sufficient details to show how the lactone isomers can be separated on scale. They are VERY close running and to separate >100g of crude material is a VERY challenging task. I urge the authors to show the TLCs of the lactone mixture and the column fractions that were collected and to give more details of how they do the column - dimensions of column, eluent/gradient, flow rate, size of fractions, etc.

The details in purification of the lactone are updated in the revised SI, including dimensions of column, brand and type of silica gel, eluent/gradient, flow rate, size of fractions and TLCs of fractions.

The Johnson-Claisen rearrangement reference (62) should be given where the Johnson-Claisen rearrangement is discussed, not later in the paper.

Johnson-Claisen rearrangement reference (now ref. 61), now moved to where the Johnson-Claisen rearrangement is first discussed.

After that, accept paper. Rest is fine.